# Bayesian networks elucidate complex genomic landscapes in cancer

Nicos Angelopoulos [1,3✉], Aikaterini Chatzipli[1], Jyoti Nangalia [1], Francesco Maura [2] & Peter J. Campbell [1]

Bayesian networks (BNs) are disciplined, explainable Artificial Intelligence models that can describe structured joint probability spaces. In the context of understanding complex relations between a number of variables in biological settings, they can be constructed from observed data and can provide a guiding, graphical tool in exploring such relations. Here we propose BNs for elucidating the relations between driver events in large cancer genomic datasets. We present a methodology that is specifically tailored to biologists and clinicians as they are the main producers of such datasets. We achieve this by using an optimal BN learning algorithm based on well established likelihood functions and by utilising just two tuning parameters, both of which are easy to set and have intuitive readings. To enhance value to clinicians, we introduce (a) the use of heatmaps for families in each network, and (b) visualising pairwise co-occurrence statistics on the network. For binary data, an optional step of fitting logic gates can be employed. We show how our methodology enhances pairwise testing and how biologists and clinicians can use BNs for discussing the main relations among driver events in large genomic cohorts. We demonstrate the utility of our methodology by applying it to 5 cancer datasets revealing complex genomic landscapes. Our networks identify central patterns in all datasets including a central 4-way mutual exclusivity between *HDR*, *t(4,14)*, *t(11,14)* and *t(14,16)* in myeloma, and a 3-way mutual exclusivity of three major players: *CALR*, *JAK2* and *MPL*, in myeloproliferative neoplasms. These analyses demonstrate that our methodology can play a central role in the study of large genomic cancer datasets.

[1] The Cancer, Ageing and Somatic Mutation Programme, Wellcome Sanger Institute, Hinxton, Cambridgeshire CB10 1SA, UK. [2] Myeloma Program, Sylvester Comprehensive Cancer Center, University of Miami, Miami, FL, USA. [3] Present address: Systems Immunity Research Institute, Medical School, Cardiff University, Cardiff CF14 4XN, UK. ✉email: angelopoulosn@cardiff.ac.uk

Cancer genomes carry, on average, four coding driver mutations[1] and typically a number of larger scale genomic aberrations such as deletions and translocations. Deciphering recurrent patterns in the mutational landscape of each cancer type can lead to better understanding of the underlying biology, increase the opportunities for targeted therapeutic interventions and to better cancer sub-typing in the absence of phenotypic evidence.

Typically, patterns of mutual exclusivity and co-occurrence are sought pairwise among driver events in order to further investigate the biology of the disease. Possible biological explanations for the former are pathway redundancy, such as the mutual exclusivity of mutations on drivers in the *RAS* pathway (*BRAF* and *KRAS*) in colorectal cancer[2] and the *RTK/RAS/PI3K* pathway in hepatocellular carcinoma (with 22–37% of patients presenting at least one alteration)[3,4]. Alternative biology at play might be in the form of non-viability of cells with mutations on the exclusively mutated drivers. Co-occurring events such as mutated *VHL* and translocations on both chromosomes 3 and 5 often seen in clear cell renal cell carcinoma[5], pinpoint events which are required in tandem.

Current approaches to elucidating such relational patterns concentrate on pairwise testing of events either via novel algorithms[6], or via statistical testing based on established tests, such as the Fisher's exact test[7]. These methods fail to consider n-way interactions and only look at cohort-wide events. As an example of the former, consider the scenario were in addition to *BRAF* and *KRAS*, *AKT1* might also be an alternative mutation in the *RAS* pathway in colorectal cancers. Due to limited statistical power, *AKT1*'s mutual exclusivity with *BRAF* and *KRAS* might not be revealed when tested pairwise with each gene separately. While, when testing the three-way interaction it might be clear that a mutation in any of the three genes is sufficient to confer a cancerous advantage to cells. Furthermore, pairwise tests only test cohort-wide events. What if a certain cancer type (or sub-type) requires either a mutation on *TP53* or a mutation on any of the three *RAS* proteins mentioned above in conjunction with a mutation to a gene downstream of the *RAS* pathway?

Mutations on oncogenes and tumour suppressor genes along with more complex genomic events can be viewed as binary variables across a cohort of patients. Thus a genomic investigation of a specific cancer type or sub-type can be viewed as a binary table of data, with rows identifying patients and genomic features of interest as columns. Data cells take binary values that indicate whether a specific patient has a known cancerous mutation in a particular, disease specific, driver gene or more generally a driver genomic event.

Genomic events to include in such studies depend on the type of cancer. In AML, for instance, events of interest might include mutations to *NRAS* and the deletion of the *5q* chromosomal arm[8]. In the context of cancer, binary events correspond well to the idea of driver events each of which might be either present or absent in a specific sample within the cohort. This is a standard approach in large cancer genomic cohorts[7–12] where cancer biologists are interested to find patterns of biological interest.

We propose the use of well-studied statistical models for the discovery of rich structural dependencies from cancer genomic datasets. Bayesian networks (BNs) are a class of complex statistical models that is central to modern, explainable Artificial Intelligence (AI) research. A BN is comprised of (a) an acyclic graph representing the conditional relationships among a set of variables and (b) a number of probability tables that detail the dependency relationships between immediate families in the graph. BNs have been proposed as disciplined models for modelling biological datasets[13]. They have been applied to the analysis for signalling data[14,15], to the construction of chromatin

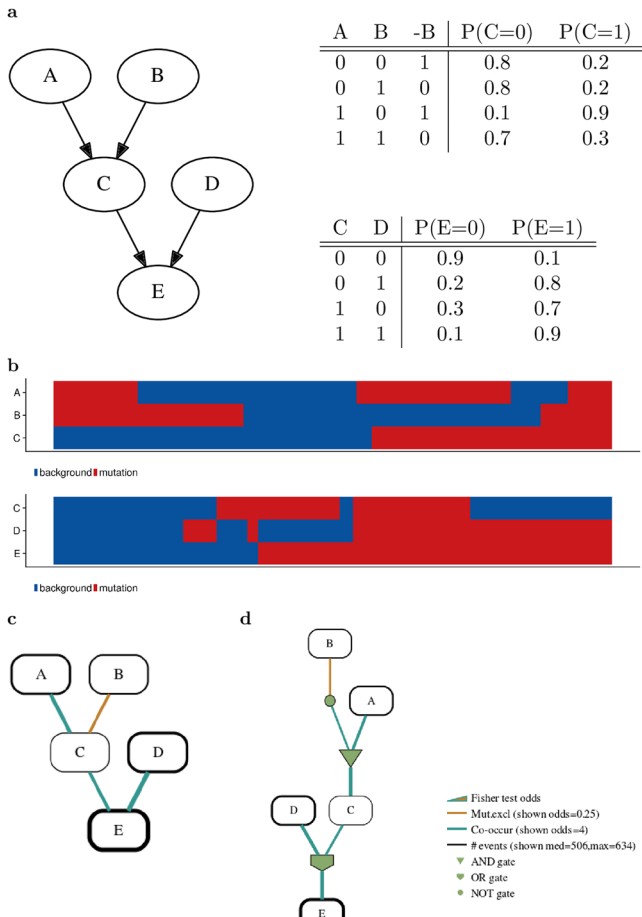

**Fig. 1 Synthetic experiment for Bayesian networks learning. a** The BN from which data are sampled. **b** Heatmaps of the sampled data that are used as input data to the BN learning algorithm. **c** The BN as learnt by *Gobnilp*. **d** The learned BN is fitted with logic gates on each of the BN families.

interaction models[16] and in restricted form, to reconstruct complex multi-generational pedigrees[17]. Learning the graph structure of a BN is a well-researched topic and a state of the art optimisation software exists that can construct optimal BN structures from data[18]. Compared to pairwise methods BNs provide complex statistical models that intuitively presents the major relations within these type of data in one model. In contrast to neural networks and deep learning methods BNs provide visual cues to which relations between our variables are important.

Although a specific algorithm (*Gobnilp*) is used in this paper, of course other BN algorithms can be used. Bayesian networks have also been used in a number of varied cancer and genomic settings. For example Wang et al. reconstruct regulatory networks using BNs[19], Rodin et al. build networks from flow cytometry data in the context of cancer immunotherapy[20] and Howey et al. use BN in Mendelian randomization in the context of genetic epidemiology[21]. The related literature includes conjunctive BNs[22,23] which were originally introduced in the context of HIV but also proposed for cancer, direct learning of logic formulae[24] and BNs for fitness landscapes[25]. In contrast to these approaches we focus on making BN learning approachable to clinicians by proposing a tight analysis framework, with only two parameters that are intuitive to set. The BN learning problem is optimally solved using a well-understood algorithm (*Gobnilp*). A number of visual cues are then employed to guide non-experts towards a

better understanding of the static relations existing in their datasets. BNs in the context of cancer progression models are implemented in *TRONCO*[26]. In these publications, interested readers can find alternative BN learning algorithms and more information on the nomenclature of BNs and their general use in computational biology.

A substantial hurdle to the establishment of BNs as a major analysis tool has been their interpretation, whereas another hurdle has been the need to select a number of parameters. The complexity of a connected network can be overwhelming when compared to pairwise tests. Here, we promote family based heatmaps as an intuitive localised viewport to the global BN structure. Heatmaps are a staple in biological data analysis used as visual means of presenting the raw data. Furthermore, we expand on the theme of interpretability by exploiting the binary nature of our datasets. Logic gates can be fitted as an additional, optional step, to further summarise and highlight the fundamental nature of BN family relations, by mapping family relations to the well understood logical operations: *AND*, *OR*, *XOR* and *NOT*. The use of an optimal algorithm, with its default parameters, allows us to reduce the number of parameters that need tuning to two: $\mu$ and $\epsilon$ which control the number of variables and density of the network respectively. Both are easy to set and have intuitive readings.

## Results

We apply our proposed methodology to a number of cancer datasets including haematological (AML, MPN, Myeloma) and solid tumours (Colon Adenocarcinoma and Glioblastoma).

**AML.** Acute Myeloid Leukemia (AML) mutational landscape analysis has previously revealed three very distinct molecular subgroups of patients reflecting distinct paths of AML evolution for prognosis stratification and disease classification in a total of 1540 patients from three clinical trials[9,27]. The structure of driver mutations allowed the unveiling of non-overlapping subgroups of patients and their fully genomic classification of AML cancer. This large and comprehensive AML study identified a number of gene–gene interactions that were to the best of our knowledge not previously known. Using the same dataset produced by Papaemmanuil et al.[27] our BN analysis elegantly captured most of the interactions reported in the study across the 11 genomic subgroups. This dataset includes mutations and selected copy number variations.

We use the AML dataset as a central vehicle for following the various steps of our methodology. Figure 2a presents the BN as learnt by *Gobnilp* while the heatmaps for a number of the families in the learnt BN are in Fig. 2b. Figure 3 shows the BN with edges coloured according to pairwise Fisher tests and Fig. 4 shows the gated BN for the same dataset.

The effect of varying $\mu$ on the number of variables and network size is shown in Supplementary Fig. 1. A plot of the variation of the size of the learnt network size when $\epsilon$ ranges from 1 to 20 is in Supplementary Fig. 3 and the distribution of events across samples of the AML cohort is shown in Supplementary Figs. 4 and 5. Finally, heatmaps for all the families in the BN learnt for this dataset are shown in Supplementary Fig. 14.

The AML subgroup identified by Papaemmanuil et al.[27] as the *NPM1* mutation genomic group (418 patients, 27%) matches well with the co-mutation patterns in our network: *NPM1*, *DNMT3A*, *FLT*3,[ITD] *TET2* and *PTPN11*. There were 275 (18%) samples with mutated chromatin and RNA splicing[27]. The most frequently co-mutated genes within this subgroup were *RUNX1*, with *SRSF2*, and *ASXL1* and *STAG2* with *TET2*, while *DNMT3A* was mutually exclusive with *STAG2*. The BN analysis was also able to identify interactions with structural variants in the AML study.

The AML subgroup with *TP53* mutations and chromosomal aneuploidy ($n = 199$, 13%) shows direct co-mutation with the *complex* karyotype, *TP53*, *-5/5q* arm and *+8/8q*. Moreover, when looking at co-mutation patterns in recurrent triplets restricted to over four observations as reported in the study, *NPM1*, *FLT3* and *DNMT3A* co-mutation pattern occurred 130 times being the top recurrent triplet, was also identified by our analysis. The same was true for *Complex*, *minus5_5q* and *TP53* (60), *Complex*, *mono17_17px_abn17p* and *minus5_5q* (49), and *Complex*, *mono17_17p_abn17p* and *TP53* (44).

In comparison to the BN network presented here, the network that is built from pairwise comparisons across all events based on Fisher statistic, Supplementary Fig. 12, is much denser and impossible to interpret. Edges in the Fisher network are drawn between events for which the Fisher statistic finds a significant correlation.

**MPN.** Genomic characterisation of 2035 patients with myeloproliferative neoplasms[28] was performed by Grinfeld et al.[10] in order to study the potential for diagnosis, risk stratification and treatment. Analysis on this cohort identified a number of genomic subgroups in myeloproliferative neoplasms, using patterns of mutually exclusive or co-mutated genes. One of the reported genomic subgroups was identified by *TP53* mutations, co-occurring with aberrations at chromosome *17p* and deletions at chromosome *5q*[29]. The network our methods constructed for this dataset is shown in Fig. 5a. The gated BN version of the same network is presented in Supplementary Fig. 6 and heatmaps for all families are shown in Supplementary Fig. 15 (within Supplementary Note 6: Familial heatmaps).

*TP53* mutations have been shown to be acquired later in disease but dominate the genomic and clinical features of these patients regardless of the initial driver of the myeloproliferative neoplasm. Another subgroup was enriched for patients with myelofibrosis (odds ratio, 6.5; 95% CI, 4.9–8.7; $P < 0.001$) and it was defined by LOH at chromosome *4q*, aberrations in chromosomes *7* and *7q* occurring together with mutations in at least 14 myeloid cancer genes (*EZH2*, *IDH1*, *IDH2*, *ASXL1*, *PHF6*, *CUX1*, *ZRSR2*, *SRSF2*, *U2AF1*, *KRAS*, *NRAS*, *GNAS*, *CBL*, *Chr7/ 7qLOH*, *Chr4qLOH*, *RUNX1*, *STAG2* and *BCOR*), accounting for the biggest co-mutation pattern in the Bayesian network.

*JAK*2 was reported as the most mutated gene in the network, framed in bold black oval, together with *CALR* and *MPL* mutations showing patterns of mutual exclusivity, confirming the functional redundancy in their pathological mechanisms. The three genes (*JAK2*, *CARL* and *MPL*) accounted for 1831 driver mutations out of the 2906 total driver mutations in the dataset. Our BN elegantly captures the three-way mutual exclusivity between these central players.

Mutations in *PTPN11*, *SH2B3*, *PHF6* and other genes not connected in the network showed a low number of mutations. For example, mutations in *MLL3* were detected in 20 patients (1.0%) and were predominantly nonsense or frameshift mainly reported in patients with acute myeloid leukemia. Our BN highlights a number of interesting co-occurrence chains, such as: *ZRSR2-GNAS-NRAS-STAG2*, *EZH2-ASXL1-CBL-C11* and *C8-C9-TET2-C4*.

Our network super-imposes well over the cluster analysis of the original paper[10]. Neighbourhoods of the network connect events that correspond to single clusters identified by the cluster analysis in Grinfeld et al.[10].

**Myeloma.** The CoMMpass data were generated as part of the Multiple Myeloma Research Foundation's Personalized Medicine Initiative, (https://themmrf.org/). Based on the Bayesian network analysis shown in Fig. 6 we were able to identify numerous

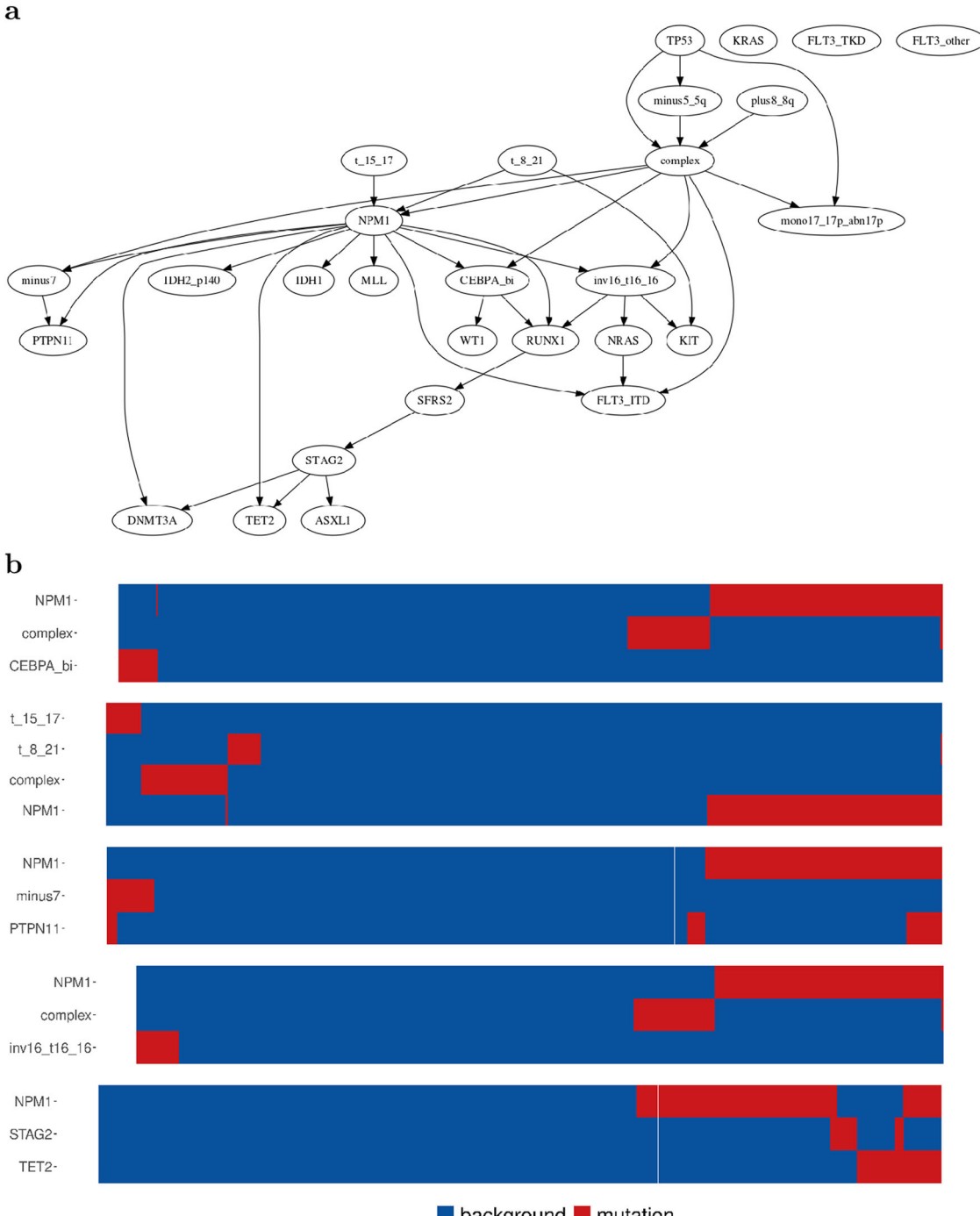

**Fig. 2 Bayesian network learnt for AML patient data. a** AML Bayesian network learned with *Gobnilp* ($\mu = 60$, $\epsilon = 7$). This is the vanilla output of the learning algorithm. **b** Family heatmaps. The complex probabilistic relationships within a Bayesian network can be broken to a number of more easily understood units. We form such units, also called families in BNs, from each node and all nodes from which an arrow points to this single node. We use heatmaps to easily communicate these relationships. Here a number of family heatmaps from the AML Bayesian network ($\mu = 60$, $\epsilon = 7$) are shown. For example, the first heatmap shows family *NPM1-complex-CEBPA_bi*. Blue plots lack of driver event while red shows presence of the event. Patients are plotted on the *x*-axis. Thus the red cluster on the bottom left of the top heatmap plots a number of patients that have the *CEBPA_bi* event but do not have the *complex* event.

dependencies of driver events in a large series of multiple mye-lomas enroled within the CoMMpass trial ($n = 724$). Specifically, the Bayesian network confirmed the known mutually exclusive pattern between the *IGH* translocations, *t(11;14)(CCND1;IGH)*, *t(4;14)(MMSET;IGH)* and *t(14;16)(IGH;MAF)* and hyperdiploid cytogenetic status, as well as the co-occurrence of *13q* deletion with *t(4;14) (MMSET;IGH)* and *1q* gain[30–32]. The gated version of the BN is shown in Supplementary Fig. 7, whereas all familial

heatmaps are given in Supplementary Fig. 16. The relationship between *HDR* and the translocations is a strong one and survives much higher $\epsilon$ values which lead to sparser networks (Supplementary Fig. 8). The network identifies mutual exclusivity between the *MAPK* pathway genes *NRAS* and *KRAS*. In our dataset, 42% of the samples contain at least one of the two mutations with only 22 (3%) of the samples containing driver mutations to both genes.

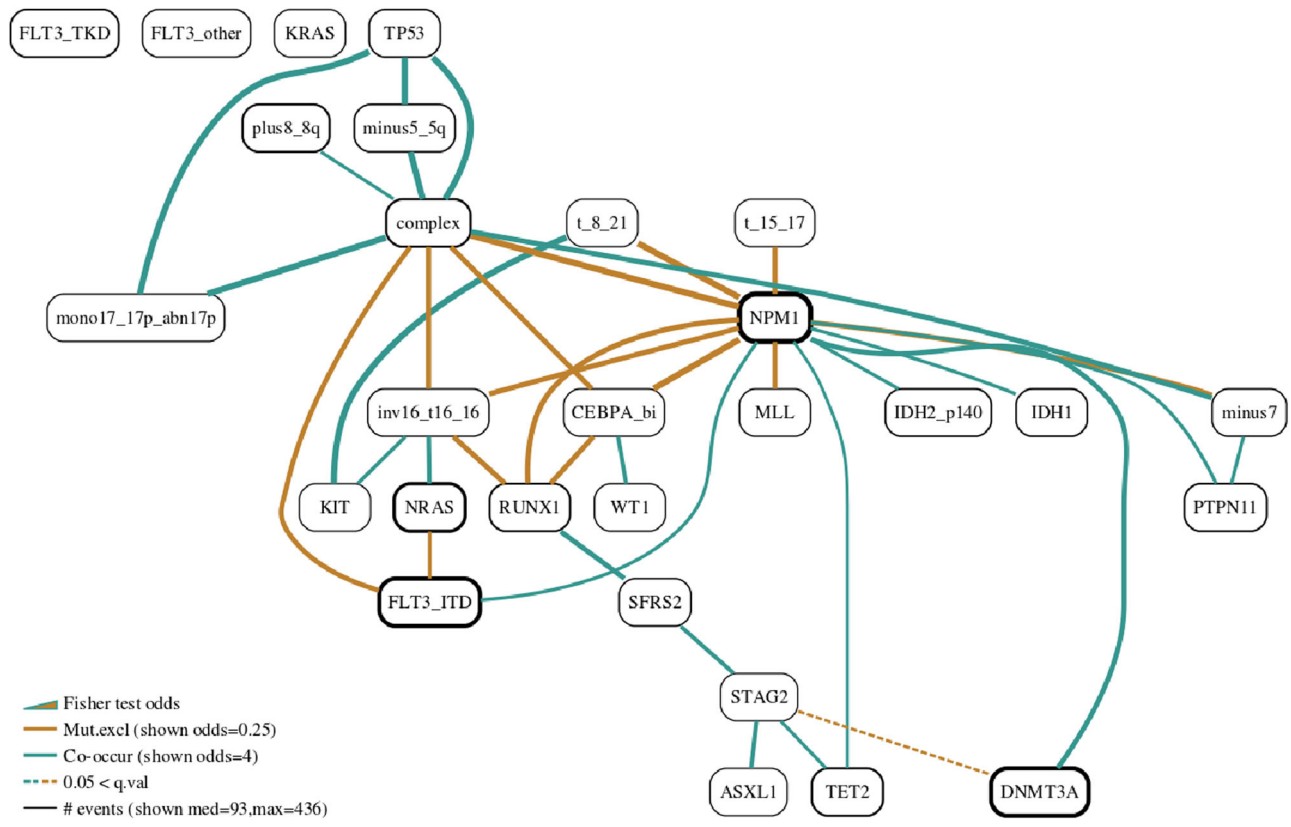

**Fig. 3 AML Bayesian network with edges visualised based on Fisher's exact test.** Edges with Fisher's exact test colours and width. To help convey more information in a succinct way, we colour edges according to the direction of the exact test result among the two mutational vectors connected by the edge. Also, the width is proportional to the odds value of the same test. It is worth noting that these quantities are not derived from the network construction algorithm, but from pairwise testing. Fisher test edges are shown for the AML Bayesian network ($\mu = 60$, $\epsilon = 7$).

Furthermore, patterns of co-occurrence were observed between other events such as between deletions of *CDKN2C* (*delCDKN2C*) and *FAM46C* (*delFAM46C*) and between *del13q14* and both *TRAF3* deletion (*delTRAF3*) and *(4;14)(MMSET;IGH)*. Interestingly, the network was also able to identify some potential dependencies of low frequent driver events such as the mutually exclusive pattern between *MAX* mutations and hyperdiploid cytogenetic status.

Finally, as expected, several gene bi-allelic inactivations were identified, such as *del17p13* and *TP53* mutation, *del13q14* and *DIS3*, *del17p13* deletion and *CYLD* mutation and *14q* deletion and *TRAF3* deletion. Overall, this highlighted the ability of the BN to identify both simple and complex patterns of abundant and rare driver events. The dataset analysed here was previously analysed by Maura et al.[11] where a simpler version of our BN was presented and described.

**Colon Adenocarcinoma, TCGA.** The Cancer Genome Atlas[33] (TCGA) project for the characterisation of human colorectal cancer originally examined 276 samples. Here we reanalysed these and additional TCGA data in order to capture colorectal cancer mutational relationship patterns. The Bayesian network in Fig. 7 was built by restricting input to known cancer driver genes. It reveals a number of expected gene relationships, such as the mutual exclusivity patterns among *BRAF*, *KRAS* and *NRAS* genes[34–36], as well as between *TP53* and *PIK3CA*[37–39]. Our network highlights the high mutation frequency of *TP53* and *APC*[40,41] presented in black bold ovals (Fig. 7). The gated BN version of the same network is presented in Supplementary Fig. 9 and heatmaps for all families are shown in Supplementary Fig. 17.

*SOX9* was mutated in a mutual exclusivity pattern to *TP53* while *MAP2K4* significantly co-mutated with *PCBP1*, *ATM* and *PTEN*. In addition, the analysis showed patterns of *ARID1A* co-mutation with *TCF7L2*, *ACVR2A* and *PIK3R1*, as well as co-mutation of *ACVR2A* with *PTEN*, *ARID1A*, *FBXW7* and *TCF7L2*. *CTNNB1* mutations were mutually exclusive of *APC*. *SMAD2* was mutated together with *CTNNB1* as expected, while due to a modest number of mutations there was a single connection from *SMAD4* in the network at this granularity (to *PIK3CA*). *SMAD4* was one of the 24 significantly mutated genes in the non-hypermutated tumours of the TCGA study. The BN analysis encapsulates the majority of colorectal TCGA while it also highlights interesting hypotheses of relationships, particularly among multiple genes, such as the co-occurrence pattern in family *ATM-PTEN-MAP2K4* and the mutual exclusivity patterns in families *RNF43-SOX9-TP53* and *CTNNB1-RNF43-APC*.

**Glioblastoma.** We next analysed data generated by the EORTC Study[12]. The molecular data were used to build a probabilistic directed graphical model that maps the co-occurrences and mutual exclusivity relationships of common glioblastoma driver genes. The BN, gated BN and familial heatmaps generated by our methods ($\mu = 5$, $\epsilon = 1$) are shown in Supplementary Figs. 10, 11 and 18 respectively.

Glioblastomas (GBMs) are the most common and most aggressive subtype of glial brain tumours[42,43]. Hundred and eighty-six pairs of primary-recurrent GBM samples of patients receiving chemo-irradiation with temozolomide were collected and ≈300 cancer genes were sequenced[12]. The most frequent genetic changes identified were mutations in *TP53*, *PTEN*, *EGFR*, *NF1*, *RB1*

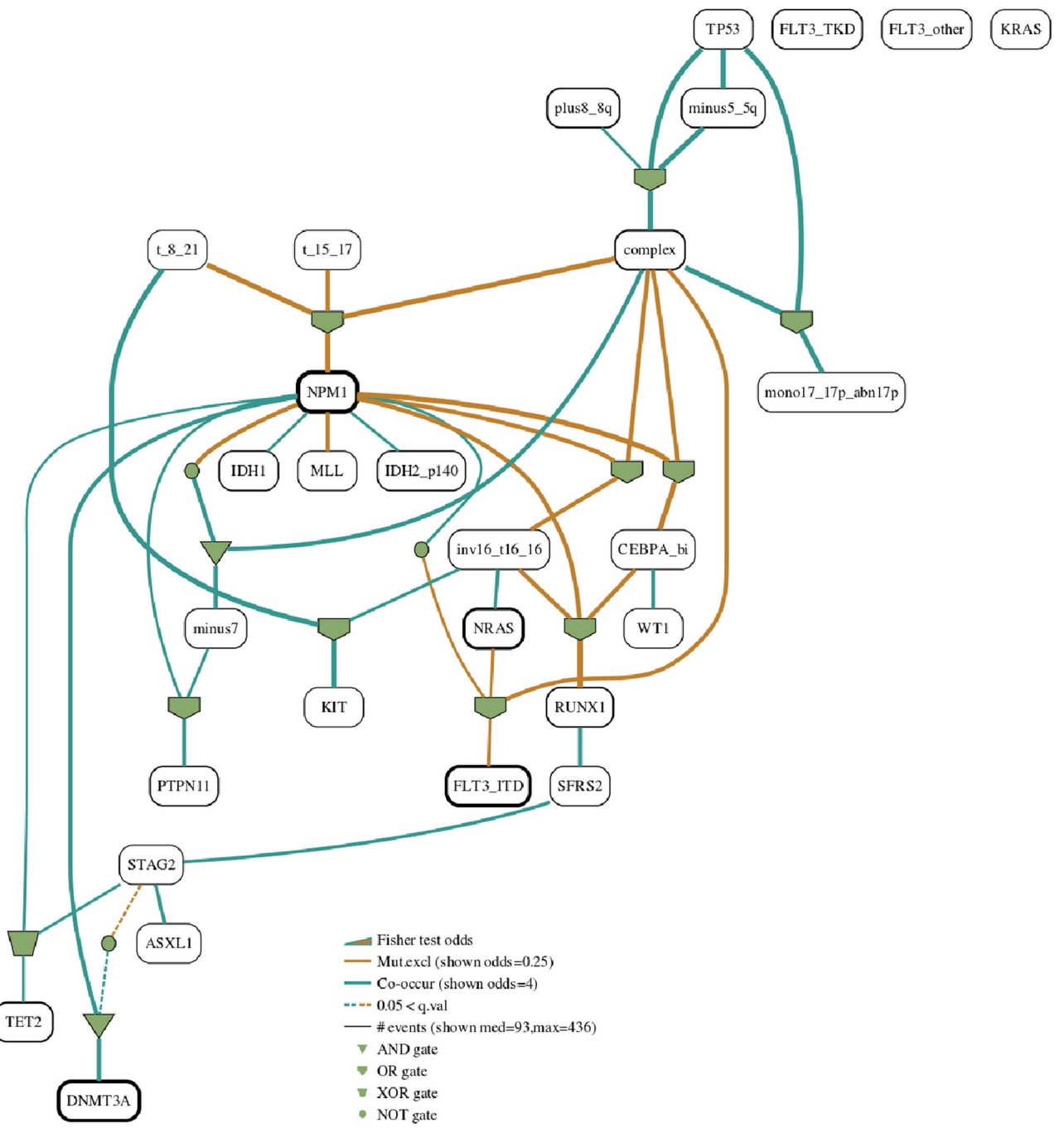

**Fig. 4 Gated Bayesian network for the AML dataset.** For each family in a Bayesian network we can fit logic gates that further help interpretation of the relationships in the network. The best configuration is selected using the Fisher statistic. Here the logic gates enhanced Bayesian network for the AML dataset ($\mu = 60$, $\epsilon = 7$) is shown.

and *PIK3CA*. Approximately 50% of tumours expressed the *EGFR*, *EGFRvIII* mutation. Our Bayesian network analysis, using mutation data only, showed expected co-occurrence and mutual exclusivity patterns of important driver glioma genes[44]. *IDH1* was significantly co-mutated with *ATRX* and *TP53* while being mutual exclusive with *TERT* promoter and *PTEN* mutations. *TP53* also co-occurred with *RB1*, while *NF1* mutations were significantly exclusive of *EGFR*. *MMR* repair gene *MSH6* co-occurred with *POLE*, which encodes a sub-unit of DNA polymerase.

In comparison to the BN network presented here the network that is built from pairwise comparisons across all events based on Fisher statistic, Supplementary Fig. 13, is much sparser. Edges in

the Fisher network are drawn between events for which the Fisher statistic finds a significant correlation.

## Discussion

We present a methodology based on multi-disciplinary work that brings a class of explainable models to the analysis of genomic and other datasets. Although the analyses shown are exclusively applied to cancer, the methodology is applicable to other diseases. Both in diseases with complex genomic elements that evolve over time (such as neurological diseases present predominantly on aging populations), but also in diseases where a number of non-genomic clinical variables can be used to build a network of

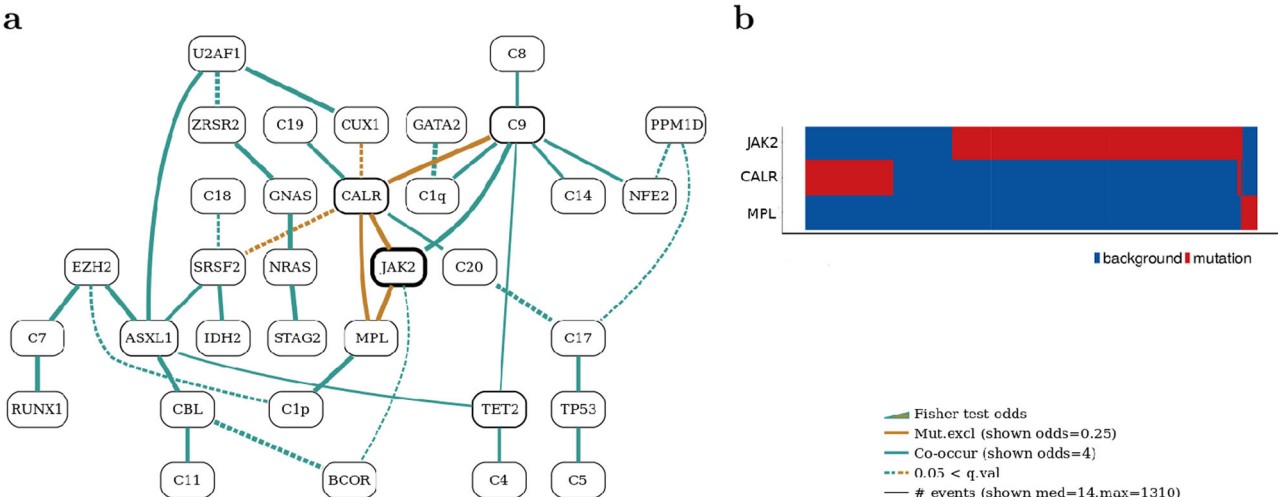

**Fig. 5 Bayesian network for the MPN dataset. a** Bayesian network constructed for the MPN dataset ($\mu = 5$, $\epsilon = 3$) along with **b** the heatmap for the *MPL* family. The heatmap plots presence (red) or absence (blue) of driver events for the three members of the family (*CALR, JAK2* and *MPL*) on the y-axis against patients on the x-axis. The heatmap, thus provides a visual aid in establishing the type of relation between these genes in the dataset. The BN captures the mutual exclusivity relationship of the three major players accounting for the majority of cases: *CALR, JAK2* and *MPL*.

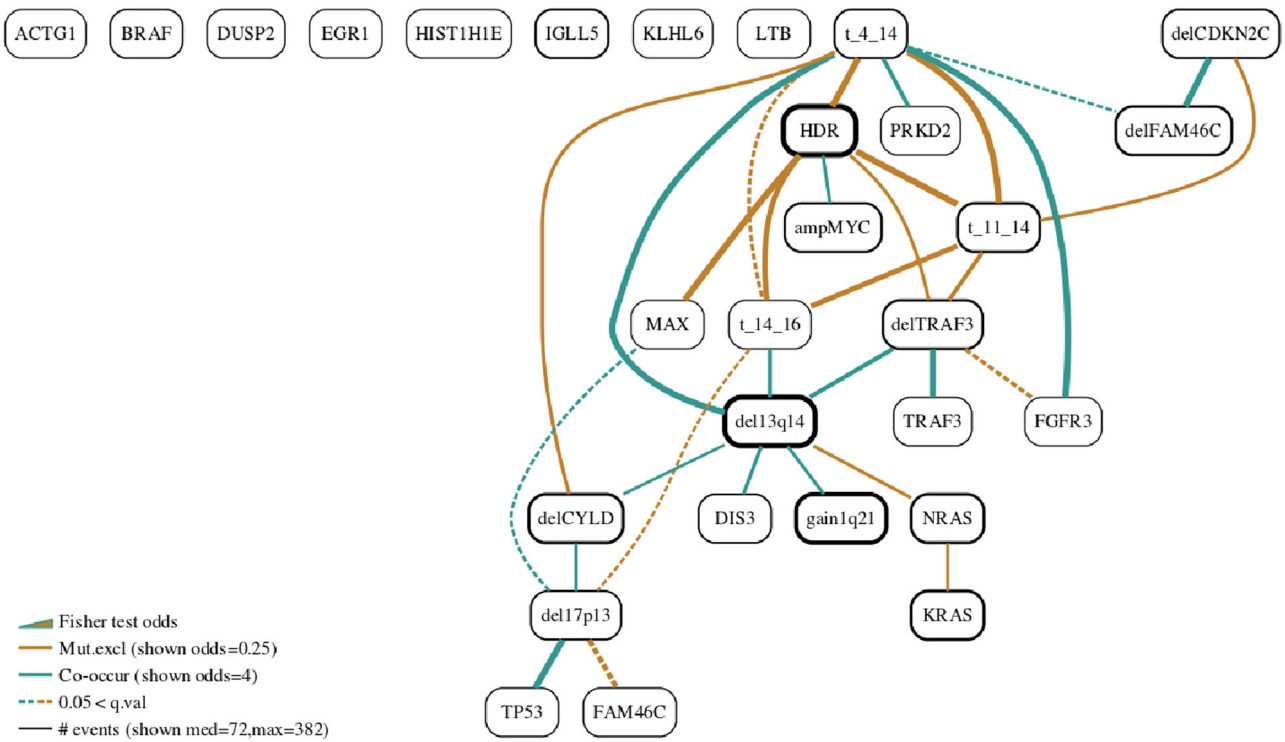

**Fig. 6 Bayesian network for the myeloma dataset.** The analysis identified a crucial four-way mutual exclusivity pattern between *HDR, t(4,14), t(11,14)* and *t(14,16)*, highlighting the central role of the three translocations and the HDR characteristic in myeloma ($\mu = 20$, $\epsilon = 3$).

dependencies (such as heart disease models based on lab and imaging variables).

A main objective of this work has been to engage clinicians and biologists in using BNs directly. BNs are an important tool that can be regularly used in the analysis of complex multivariate clinical datasets. In order to make these models more accessible to experimentalists we have married an optimal BN learning algorithm with a number of interpretation cues. Furthermore, our results on specific datasets demonstrate the power of the methodology. Crucial to this mission has been the ability to restrict the number of parameters to two easily understood parameters. Thus allowing unmediated control to non specialists.

The presented methodology empowers clinicians in a path from classical pairwise testing towards more complex statistical models. As the generation of big genomic datasets is becoming routine, complex models have a positive role to play in uncovering the intricate relationships between driver events. The limiting factor is often that the scientists generating the data have no previous experience in dealing with such formalisms. This research brings this important class of models within their grasp.

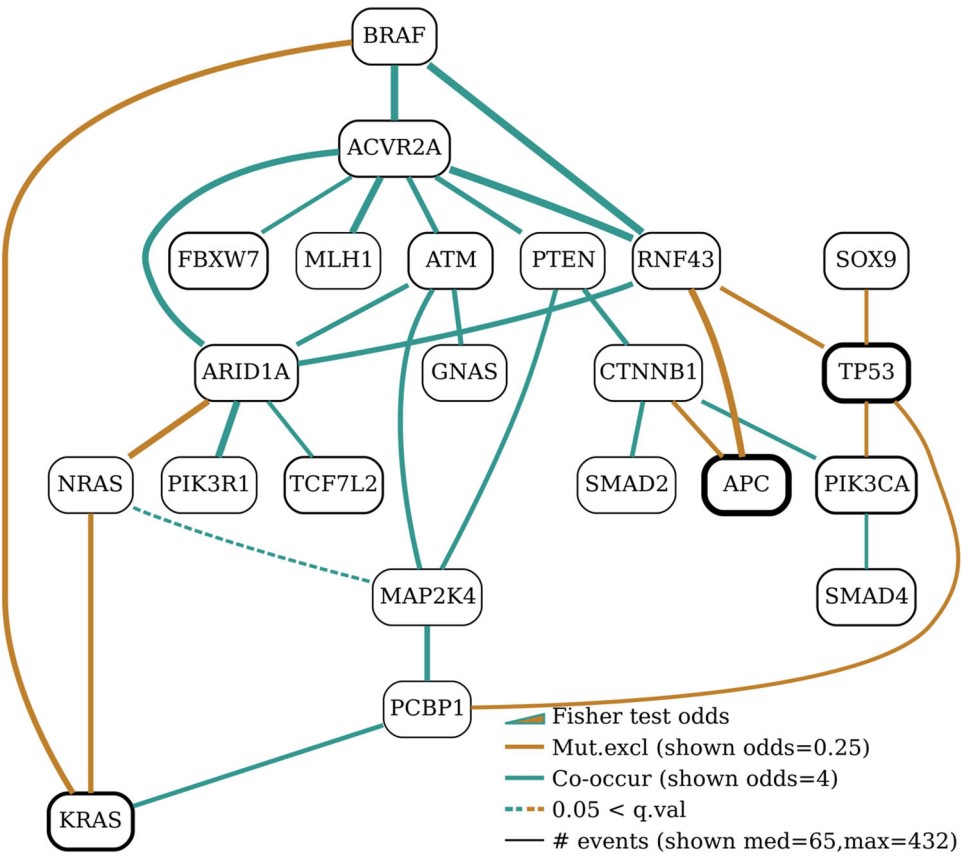

**Fig. 7 Bayesian network for the colon adenocarcinoma data from TCGA.** Highlights include a consistent mutual exclusivity pattern among *BRAF*, *KRAS* and *NRAS* and a co-occurrence pattern among a number of drivers with a small number of individual mutations (shown linked to *ACVR2A*). The parameters used in learning this network were $\mu = 5$ and $\epsilon = 1$.

We applied the proposed methodology to five large scale, cancer genomic datasets to reveal patterns of complex relationships that were either previously unknown or had been characterised in non systematic ways. We have chosen datasets from a wide variety of cancers to ensure our methods are widely applicable. Crucially, the selected datasets contain all genomic aberrations that are considered to play a driving role in the specific cancer type. Clinicians and biologists that have a detailed knowledge of specific cancer types, are the target audience of the proposed methodology. We provide a few parameter settings so as to avoid reliance on technical statistical knowledge. Furthermore, in all the datasets analysed it was always clear that the BN families highlighted strong links that can be reported confidently. Parameters, $\epsilon$ and $\mu$, play a supporting role and are about striking the right balance between sparse and dense networks when communicating the findings of each study. Our networks, consistently across the cancer types, are able to confirm a number of well-established interactions while in addition they propose a number of interactions that warrant further investigation.

## Methods

**Bayesian networks**. The graph structure of a BN provides a way of decomposing the joint probability of its variables. For the BN in Fig. 1a the structure dictates that:

$$p(A, B, C, D, E) = p(C|A, B)p(E|C, D)p(A)p(B)p(D) \qquad (1)$$

For each conditional relation defined by the structure of a BN, there exists an associated probability table that assigns a probability distribution over the values of the dependant variable given each possible combination of its antecedents. For example, the BN in Fig. 1a defines $p(C = 0|A = 0, B = 0) = 0.8$ and $p(C = 1|A = 0, B = 0) = 0.2$. This network can be decomposed into two families:

one for $C$ ($A$-$B$-$C$) and one for $E$ ($C$-$D$-$E$). Although probabilistic in nature, the example BN encodes 2 logical relations: $C \approx A$ AND $\neg B$ and $E \approx C$ OR $D$.

BN structure learning methods reconstruct the graph part of a BN from data. Score-based algorithms search the space of possible graphs to find those that score highly on the given data. Although finding the single best graph is in general an intractable problem (NP-hard) the use of constraints within an optimisation framework can routinely solve the size of the problems we are interested in and which rarely contain more than 50 variables. Specifically, we use throughout, the *Gobnilp* software[18] for learning the basic BN structure. This software maps the problem into an integer linear programming formulation which can be tackled by optimisation programs such as SCIP[45]. Using this strategy, *Gobnilp* is able to find the optimal network explaining the input data. ILP guarantees that if the algorithm terminates then the optimal solution has been reached. Depending on the size of the problem ILP might fail to terminate within a reasonable amount of time. For the size of problems analysed here, this has not been an issue. All our examples can be reproduced on computers with very modest capabilities (Supplementary Note 1: Software).

Figure 1 shows the results of a small in silico experiment. 1000 points are sampled from the joint distribution defined by the BN in Fig. 1a. Using these data points the BN structure is recovered in full (Fig. 1c). A version of the learned network with fitted logic gates is shown in Fig. 1d. Figure 2a shows the BN constructed by running *Gobnilp* on the AML data set[9] as input.

As we (a) do not have intervention experiments to pinpoint causality and direction of arrows, and (b) we take an approach that is centred around families in deconstructing the networks, we will not visualise any arrow directions in subsequent networks. In our networks no special interpretation should be given to the directionality of edges between family members. In particular, children-parents relations should be interpreted as correlations and not as causations. The exact mutational pattern for each family is best visualised by the heatmap of the input values for the family variables (Fig. 1b). To increase the visual interpretability of the networks, we colour the edges according to pairwise exact Fisher's test. It is worth noting that this information is not derived by, or directly related to, the BN learning algorithm. Yellow colour is used for edges between mutually exclusive events (odds ratio <1) and green is used for edges between co-occurring events (odds ratio >1). Edge widths are drawn in proportion to the value of the odds ratio for the test. Dotted edges indicate a non-significant corrected *p*-value for the Fisher test between these two variables. Figure 3a shows the BN constructed for the AML

cohort with Fisher test controlling the colour, line type of edge and the width of edges.

To further assist with the interpretation of binary genomic datasets, we propose an additional, but optional step, of fitting logic gates into each network family. From all possible combinations, the set of gates that minimises the *p*-value of the Fisher test when testing the child vector to the vector resulting from applying the logic gate to the values of the parents. Figure 4 shows the AML network with fitted logic gates. We will refer to these logic gates enhanced networks as gated BNs. Whether specific researchers find the gates of help, depends on many factors, inclusive of the specifics of the dataset. We thus suggest the optional use of gates in situations where the interpretation is possible, dependant on the expertise of the investigating team. Our analysis methodology constitutes of four main steps.

**BN learning**. A binary or discrete matrix of genomic aberrations is the input to the *Gobnilp* software[18], which learns the best Bayesian network scored by a specific likelihood function. Looking for the optimal structure is mapped to an integer linear programming problem which is then solved using the SCIP optimisation suite[45].

We have used default parameters for the *Gobnilp* software throughout as we seek that our methodology is accessible to non-experts. We utilise two simple parameters for controlling the learning process. The first, $\mu$, is employed prior to running the BN learning algorithm and is a simple means of controlling the number of variables to be included in the network learning step. The parameter's purpose is to remove any variables that occur too infrequently. For example, a value of $\mu = 10$ will curtail any genomic event that appears in less than 10 samples. The rationale is that these variables will not contribute to important parts of the network. In terms of the workflow of analysing cancer datasets, clinicians would normally collect from the literature and own knowledge a large panel of driver events that may be implicated. Parameter $\mu$ is an intuitive way by which variables that do not appear in the dataset under analysis, are removed. For each of the datasets we analyse in this paper, we report the $\mu$ value used. The effect of different $\mu$ values on the number of variables and associated number of edges for the AML dataset, is discussed in Supplementary Note 2: Parameter selection and is shown in Supplementary Fig. 1. Its effect on the number of variables for the other datasets is shown in Supplementary Fig. 2. In machine learning terms $\mu$ is not a proper parameter that is fine-tuned (also known as regularised) during experimentation. Selecting a value for $\mu$ is a much simpler decision typically taken by clinicians at the beginning of data processing.

The second parameter ($\epsilon$, for edge penalty) controls the sparsity of networks. Following a Bayesian machine learning framework[46] *Gobnilp* can express the posterior of a model ($M$) given data ($D$) as $P(M|D) \propto P(M) * P(D|M)$ where $P(M)$ is the prior probability of $M$ and $P(D|M)$ is the likelihood of the data for the specific model. The usual agnostic prior is to fit a uniform distribution over $P(M)$, which means that the model with the maximum posterior probability is the one with the highest likelihood score. Given edge penalty parameter $\epsilon$, a network ($M$) with $n$ number of edges will have a prior value $P(M)$:

$$P(M) \propto e^{-n*\epsilon} \qquad (2)$$

which penalises dense networks. $\epsilon$ takes a simple integer value, typical in the range of $1-20$. Its effect on network size (number of edges) for AML dataset is shown in Supplementary Fig. 3. For each constructed network we report the value of this parameter.

Both parameters are straightforward to set and have intuitive readings. Researchers can set these to stringent levels when preparing networks for publication, whereas when using the networks for searching for validation targets they can use more permissive values. *Gobnilp* takes a number of other parameters that can control for the minimum and maximum number of parents or express conditional independence constraints. Other technical parameters can control the function that scores how well a particular BN fits the data.

We have found the generated Bayesian networks to be robust and meaningful. The latter is attested by the analyses of five well-known datasets as discussed later in the paper. Robustness of the networks can be an issue with BN learning but in the datasets we analysed we found this not to be the case. An important factor is that the number of samples far outweigh the number of variables. In these large cohort genomic studies where driver events are reasonably well established and limited in number this will always be the case. Furthermore, in each dataset we will typically run *Gobnilp* under a range of $\epsilon$ values which controls the density of networks. Robustness is also attested by the fact that in all cases denser networks incorporate the vast majority of edges of sparser ones (Supplementary Note 3: Robustness). For instance, all significant edges (shown as continuous lines) in Supplementary Fig. 8 which shows the gated BN for myeloma at $\epsilon = 12$ are also present at the much denser network for the same dataset shown in Fig. 6a ($\epsilon = 3$). More formally, we found that edges in the most sparse networks are consistently present in denser networks (99.3%, Supplementary Note 3: Robustness).

**Family heatmaps**. For each family in the constructed BN a heatmap can be drawn to guide interpretation. Families in BNs pinpoint multi-way interaction patterns in the dataset, while heatmaps precisely illustrate the mutational patterns of all participants. Heatmaps are trusted visualisations that are well understood by biologists and clinicians. The heatmaps for the two non trivial families of the example BN of

Fig. 1a are shown in Fig. 1b. Figure 2b shows the most important family heatmaps for the AML dataset. Finally, a crucial heatmap for the MPN data is shown in Fig. 5b. This heatmap demonstrates the power of visualising family relationships, clearly showing the 3-way mutual exclusivity and the extent of coverage of the three major players: *JAK2*, *CALR* and *MPL*, which together cover the vast majority of cases.

Each heatmap depicts the data in the cohort that correspond to a single family. A family is comprised of a single child node and all nodes above it from which edges are connected to that node (parents). For example, the heatmap in Fig. 5b shows the source data for family *JAK2-CALR-MPL*. *MPL* is the child node and nodes *JAK2* and *CALR* are the parents. Of course, in BN families there is no restriction to two parents. Family heatmaps plot family members on the *y*-axis and patients on *x*-axis. The presence of an event is plotted in red while absence is shown in blue. These heatmaps provide a visual cue to the kind of relationship between the family members that exists in the primary data. The heatmap in Fig. 5b clearly shows that there is a three-way mutually exclusive relation between the three drivers (*JAK2*,*CALR* and *MPL*), as patients that have mutation in one of them (red blocks), do not have a mutation in the other two drivers (blue sections at the same *x*-axis positions).

**Fisher edges**. Edges in the BNs that we display are coloured according to the odds ratio of Fisher's exact test (R function fisher.test()) between the end points of that particular edge. An odds ratio value of less than 1 indicates a mutual exclusive pattern (yellow colour) with co-occurrence corresponding to odd ratio values greater than 1 (green colour). It is worth noting that each edge is coloured according to the pairwise test and that this is done to give further visual cues of the relationships within the dataset. For example, in the family *STAG2-NPM1-DNMT3A* of Fig. 3, the edge *STAG2-DNMT3A* shows a mutual exclusivity between *STAG2* and *DNMT3A* (odds ratio value of less than 1) whereas edge *NMP1-DNMT3A* is coloured as a co-occurrence relation between *NPM1* and *DNMT3A* (odds ratio value of 1 or more).

We draw edge widths proportional to the absolute value of the odds ratio value of the test. Edges are drawn as continuous lines if multiple hypothesis corrected (by default using Benjamini and Hochberg, or any accepted by p.adjust() R function) *p*-value of the test is less than 0.05 and as dotted lines if the test was equal or bigger than 0.05. The edge *STAG2-DNMT3A* (Fig. 3) is shown as a dotted line as the *p*-value is not significant, whereas the edge *NMP1-DNMT3A* in the same BN is drawn in continuous pen as the *p*-value is significant. The width of edges is proportionate to Fisher odds value for the pair of events connected by the edge. The width on the legend is calibrated to the value displayed on the same line. In Fig. 3 the co-occurrence width displayed at the legend corresponds to an odds value of 4.

**Family gates**. For each family we fit all possible logic gate configurations choosing the combination minimising the *p*-value of the Fisher test between the child values and the vector resulting from applying the logical formula to the parents' vectors. The fitting of logical gates respects the conditional independencies encoded in the learned Bayesian networks. For instance, for the family *complex-NPM1-CEBPA_bi* in Fig. 3, the *OR* gate is fitted between the edges *complex-CEBPA_bi* and *NPM1-CEBPA_bi* already identified by the BN learning algorithm (gated version is shown in Fig. 4).

The user can easily choose which gates should be considered from the following: *AND*, *OR*, *NOT* and *XOR*. *XOR* is given a preference over *OR* when they derive identical test statistics. By default all gates are considered apart from *NOT*- which can be added if required. The software applies straightforward multi-input simplifications, where $((A \ AND \ B) \ AND \ C) = D$ is shown as a 3 input, 1 output single gate $AND(A, B, C) = D$.

**Statistics and reproducibility**. The BN learning results can be reproduced by running *Gobnilp* with default parameters apart from those we explicitly set, on the data provided in Supplementary Data 1: Input datasets.

The only statistical test we utilise is Fisher's exact test. The test is used to highlight pairwise relationships in the constructed Bayesian networks. Edges with significant *p* value (<0.05) are shown as continuous lines whereas non-significant edges are shown as dotted lines. The same test is also used to choose between all possible logic gates. In this case, the gate with the minimal value is selected.

**Genomic datasets**. Cancer and normal samples were sequenced using a custom-designed targeted cancer panels as described below for each study. Sequencing was performed on Illumina HiSeq2000 machine using 75-bp paired-end method with the target of 1Gb sequence per sample. Mutation analysis was performed using in-house algorithms developed at the Welcome Sanger Institute. Sequencing reads are aligned to the NCBI-built human genome using the BWA algorithm to create a BAM file with Smith-Waterman correction with PCR duplicates removed (http://broadinstitute.github.io/picard/). Calling of substitutions was done with the Caveman algorithm[47] and insertions/deletions (indels) with Pindel[48]. All datasets used here were previously published and all had obtained informed consent.

**AML**. We use the dataset of AML patients first introduced in Papaemmanuil et al.[27], which sequenced *DNA* extracted from peripheral blood granulocytes or bone marrow mononuclear cells. Patients were drawn from three prospective multi-location clinical trials of the German-Austrian AML Study Group (AMLSG). The dataset includes a number of important recurrent alterations including fusion genes and copy number alterations. Our starting dataset contains the full set of variables analysed in the primary publications[9,27]. The AML binary matrix used for the BN learning reported there contains 1540 rows (samples) and 84 columns (events/variables). As many variables are exploratory, having low frequency, we used a stringent $\mu = 60$, which reduced the number of variables from 84 to 28. The AML BN is shown in Fig. 3 and was built with $\epsilon = 7$.

**MPN**. The MPN dataset was as reported by Grinfeld et al.[10]. Briefly, 69 myeloid driver genes in 2039 patients with myeloproliferative neoplasms were sequenced. Thirty-three genes had driver mutations in at least five patients, with mutations in one of *JAK2*, *CALR*, or *MPL* being the only driver event in 45% of the patients. In addition to driver mutations, we model a small number of high order genomic events that are known to be important in MPN. The MPN binary matrix that was used for the BN presented here included 2039 rows and 46 columns. The learnt BN is in Fig. 5 and the parameters used were $\mu = 5$ and $\epsilon = 3$. The application of $\mu$ removed five-event variables.

**Myeloma**. The BN learning algorithm was ran on a large multiple myeloma (MM) series composed by 724 patients from the COMMPASS series, an observational clinical trial founded by Multiple Myeloma Research Foundation. Only genomic features with a known driver role have been included[11]. The final binary matrix used for learning the network contained 724 rows and 29 columns. The myeloma BN is shown in Fig. 6 and was built with $\mu = 20$ and $\epsilon = 3$. The parameter value $\mu = 20$ reduced the number of event variables from 69 to 29.

**Colon adenocarcinoma, (COAD), TCGA**. To define the mutational spectrum of colorectal cancer, the TCGA consortium performed exome capture DNA sequencing on 523 tumour and normal pairs. Sequencing was performed in paired-end mode with Illumina HiSeq 2000. Illumina sequencing libraries were amplified by bridge-amplification process using Illumina HiSeq pair read cluster generation kits (TruSeq PE Cluster Kit v2.5, Illumina) according to the manufacturer's recommended protocol. BAM files of the sequenced samples were obtained and reanalysed using the CGP pipeline at the Sanger Institute. Biospecimens were collected from newly diagnosed patients with colon or rectum adenocarcinoma undergoing surgical resection and had received no prior treatment for their disease, including chemotherapy or radiotherapy. All cases were collected regardless of surgical stage or histology grade. Cases were staged according to the American Joint Committee on Cancer (AJCC) staging system. Each frozen tumour sample had a companion normal tissue possibly from blood/blood components. Our BN learning dataset contained 523 rows and 22 columns. The BN for the TCGA/COAD dataset is shown in Fig. 7. It was built with $\mu = 5$ and $\epsilon = 1$. $\mu = 5$ removed no event variables.

**Glioblastoma**. Samples from histologically proven glioblastoma patients before and after receiving chemo-irradiation with temozolomide were collected in the EORTC network (10 institutions in 6 countries, as reported by Draisma et al.[12]). Formalin-fixed paraffin-embedded tissue was processed at the Erasmus MC, Rotterdam. The study included samples from primary and recurrent 186 individuals paired against normal. A total of 300 cancer genes were sequenced[12]. Here we concentrated on known driver mutations in driver genes on post treatment samples. Genes were required to be mutated in a minimum of five samples ($\mu = 5$). Our BN learning dataset included 179 rows (samples) and 21 columns (genes). Supplementary Fig. 10 shows the BN for the glioblastoma data, built with $\mu = 5$ and $\epsilon = 1$. The application of $\mu$ removed no driver events as it is likely that the dataset was already cleaned for rare events.

**Reporting summary**. Further information on research design is available in the Nature Research Reporting Summary linked to this article.

## Data availability

All datasets analysed here were previously published elsewhere. The matrices of genomic events used as input to BN learning are included in the source code of our software (Supplementary Note 4: Datasets) and provided as Supplementary Data 1: Input datasets.

The AML dataset is available at the European Genome-Phenome Archive with accession number EGAS00001000275[49]. Here we used the version available at https://github.com/gerstung-lab/AML-multistage. The myeloma primary genomic dataset is available at the European Genome-Phenome Archive with the accession number EGAS00001001299[50]. The glioblastoma primary dataset is available from the same archive with accession number EGAD00001004593[51]. The colon adenocarcinoma data is available from the Genomic Data Commons Portal[52]. The myeloproliferative neoplasms dataset was obtained from the corresponding author of the cited paper[10]. All datasets can also be requested from the corresponding author.

## Code availability

All code and scripts used here are made available as open source at https://github.com/nicos-angelopoulos/gbn (version 0.2). The code is also available on Zenobo[53]. Our code only depends on open source and free for use in academic settings software. Because of the large number of software dependencies, we also provide a complete operating system image for the Raspberry pi 4 architecture: https://stoics.org.uk/nicos/sware/gbn/gbn_image.html. All code can also be requested from the corresponding author.

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

## Acknowledgements

N.A. is supported by the Ser Cymru II programme which is part-funded by Cardiff University and the European Regional Development Fund through the Welsh Government. F.M. is supported by the American Society of Hematology, Riney Family Foundation and the Sylvester Comprehensive Cancer Center NCI Core Grant (P30 CA). This research benefited from core institutional funding to the Wellcome Sanger Institute. The authors would like to thank James Cussens for kind help with *Gobnilp* and discussions on machine learning of graphical models.

## Author contributions

N.A. and P.J.C. conceived and designed the study; N.A., A.C. and P.J.C. analysed the data and wrote the manuscript. J.N. and F.M. provided the datasets and biological discussion for the MPN and myeloma cancers respectively. All authors approved the final manuscript.

## Competing interests

The authors declare no competing interests.
