## [Peer Review File · Communications Biology]

Reviewers' comments:

Reviewer #1 (Remarks to the Author):

The manuscript focuses on the application of Bayesian network inference in cancer. A variety of data were employed and the effects of number of variables in the network learning step as well as the sparsity of networks was investigated. Overall, the manuscript addresses an important topic and provides insight on facilitating the use of such methodologies by biologists. Some comments are given below:

- The methodology includes an integer linear programming stage for learning the BN structure. It would be beneficial to comment on the potential degeneracy of the solutions obtained.
- Default parameters are used for the optimisation with Gobnilp, justified by the ease of software use, however some comment on the effect of parameter tuning would be welcome.
- In controlling the learning process, parameter μ (number of samples that a genomic event appears), seems more akin to data preprocessing (and therefore data-dependent) rather than a parameter that can be considered as an integral part of the computational protocol.
- The comment is made that "Robustness of the networks can be an issue with BN learning but in the datasets we analysed we found this not to be the case". The authors should elaborate on how they assessed robustness.
- The effect of parameter μ is shown in Suppl. Figure 1 for the AML dataset, some comment on whether similar effects apply on the other datasets would be useful.
- Discussion of results would benefit from illustrating some key signalling events that were identified in the analysis and constitute novel observations.
- Indication of application to diseases other than cancer is mentioned in the manuscript, so the discussion could extend to some examples.

Minor comments:

- "other BN algorithms of course can be used", please rephrase to "of course other BN algorithms can be used"
- "Interested readers can find in these publications alternative BN learning algorithms and many more information in the nomenclature of BNs and their general use in computational biology", please rephrase to "In these publications, interested readers can find alternative BN learning algorithms and more information on the nomenclature of BNs and their general use in computational biology"
- "as we seek for our methodology to be accessible to non-experts", please rephrase to "as we seek that our methodology is accessible to non-experts"
- "We provide few parameter settings as to avoid reliance", please rephrase to "We provide few parameter settings so as to avoid reliance"

Reviewer #2 (Remarks to the Author):

The manuscript represents an attempt to develop and validate an AI based tool for a simple, network-based visualisation of big data variables and their relationships that would inform clinical and biological scenarios.

The authors focus on generating networks of cancer genomic data from five different blood and solid cancers.

The presented results largely vindicate their aims. However, there are several areas in each tumour type that need to be clarified further.

Methods

1. It is stated in the manuscript that the Bayesian networks are explainable AI models. What is the meaning of this and how does this become evident and verifiable in the case of the five different cancer patient cohorts studied here?
2. in p10 para 3 it is stated 'For instance all significant edges (shown as continuous lines) in

Supplementary Figure 6 which shows the gBN for myeloma at $e = 12$ are also present at the much denser network for the same dataset shown in Figure 6 ($e = 3$).

However, according to Suppl Fig 2, the density of edges anticorrelates with the e value, i.e., the density of networks in Suppl Fig 6 and Fig 6 should be the other way round.

3. p12-13 'For instance, for the family complex-NPM1-CEBPA bi in Figure 3 the AND gate is fitted between the edges complex-CEBPA bi and NPM1-CEBPA bi already identified by the BN learning algorithm (gated version is shown in Figure 4).'

The gate symbol shown in Fig 4 is OR not AND.

Results

AML

It is well established that $t(15;17)$ can co-occur with FLT3-ITD in 20-30% of cases. This is not evident in Figures 3 & 4?

Suppl Fig 15 is mentioned in p15 but not included in the Suppl File.

MPN

If JAK2, CALR and MPL are independent early drivers, what is the meaning of MPL being the child and the other two the parents in the family?

Myeloma

NRAS and KRAS mutations are found in 40% of patients and in all cytogenetic subgroups, especially translocations. If my reading of the network is correct, the only conclusion that can be drawn from Fig 6 and Suppl Fig 5 is that NRAS and KRAS mutations are mutually exclusive to each other and to $del13q$. If so, then the network would be missing important clinical and biological information.

In fact, in p20 it is stated that 'Furthermore, novel patterns of co-occurrence were observed between other recurrent events such as between mutations on the MAPK pathway (NRAS and KRAS) and hyperdiploid cytogenetic status.'

Again, this co-occurrence is not evident in the network.

Colon adenocarcinoma

p21. 'SOX9 mutations were mutated in a mutual exclusivity pattern to TP53 while significantly co-mutated with PCBP1 and MAP2K4.'

From the network it is evident that PCBP1 and MAP2K4 are co-mutated, the only link-edge for SOX9 denotes mutual exclusivity with TP3, itself in mutual exclusion with PCBP1. How does this pattern lead to the conclusion that SOX9 is co-mutated with PCBP1 and MAP2K4?

Suppl Fig 16 is missing.

Reviewer #3 (Remarks to the Author):

The paper is intended to present and popularise and simplify the use of Bayesian networks (BN) as a method for inferring driver events in large cancer genomic datasets.

The paper suggests that BNs can overcome the limitations that are inherent in inference based solely on pairwise comparisons when the underlying system is not fully characterised by pairwise interactions. This is common in biological datasets where interactions of higher arity are fundamental to the system being investigated and where pairwise comparisons can lead to incorrect conclusions or missing important information.

They propose to do this with Bayesian networks augmented with heat maps as well as an optional step to fit logic gates to the fitted model to enhance explainability. Heat maps are incredibly common in biological sciences and it is clear that this will be easier to understand than many other methods for a lot of biologists. Additionally the logic gates being fitted provides an easy to

interpret visualisation of the network which is otherwise hard to interpret.

The proposed method for building the networks is simple and only requires the optimisation of two parameters. These parameters seem reasonable and the authors explore the effect of varying these parameters in the supplementary material. They make note that many other methods would also work for building the network (there are many algorithms for this purpose) but have chosen one which is likely to be easy to use for biologists/computational biologists.

To test the method in a biological setting they attempt to recapitulate some known results which were acquired through different means than a BN. The results show that the BN manages to produce very similar results to the already published results. Judging by this the method appears to perform well. They are not presenting new biological answers but demonstrating the power of a computational technique, so the analysis and statistical information provided all seems appropriate.

Overall I believe this paper is an interesting contribution to the field, it could help to popularise the use of an important technique into the field of cancer biology and I believe it could be published in this journal.

Reviewer: 1

Dear reviewer thank you for your time in reviewing our manuscript.

- The methodology includes an integer linear programming stage for learning the BN structure. It would be beneficial to comment on the potential degeneracy of the solutions obtained.

The ILP approach used guarantees to find the optimal solution if it terminates. However, you are correct that ILP may fail to terminate, in which case a sub-optimal solution will have been reached. For the size of problems in this paper this has not been an issue at all. All our networks can be reproduced on the raspberry pi 4 computer which has the computational power of a modest smartphone. We brought forward relevant text and changed it to read as follows (page 7)

ILP guarantees that if the algorithm terminates then the optimal solution has been reached. Depending on the size of the problem ILP might fail to terminate within a reasonable amount of time. For the size of problems analysed here this has not been an issue. All our examples can be reproduced on computers with very modest capabilities (Supplementary Section 1).

- Default parameters are used for the optimisation with Gobnilp, justified by the ease of software use, however some comment on the effect of parameter tuning would be welcome.

It is always a difficult balancing on how much technical details to include when attempting to address a multi-disciplinary audience. We have explicitly set out to attract users to the technology that would not habitually use it. We feel the original text does this well. We have added the following to page 10:

Gobnilp takes a number of other parameters that can control for the minimum and maximum number of parents or express conditional independence constraints. Other technical parameters can control the function that scores how well a particular BN fits the data.

- In controlling the learning process, parameter mu (number of samples that a genomic event appears), seems more akin to data preprocessing (and therefore data-dependent) rather than a parameter that can be considered as an integral part of the computational protocol.

Parameter μ is indeed not what in ML learning would be called a tunable parameter. Please consider though, that this not a machine learning paper. A clinician that is leading a large genomic study will have to decide on a value for μ regardless of what part of the process it should be considered. We are indeed of the same opinion as the reviewer, however, we didnot want to be open to the accusation that we are hiding parameters in order to make the framework

less complex than it really is. In reality, the presented framework has a single parameter, which makes for an even more compelling case. However, as our main audience is non-ML scientists, we keeping the main exposition intact. We have added (page):

In machine learning terms μ is not a proper parameter which is fine-tuned (also known as regularised) during experimentation. Selecting a value for μ is a much simpler decision typically taken by clinicians at the beginning of data processing.

- **The comment is made that “Robustness of the networks can be an issue with BN learning but in the datasets we analysed we found this not to be the case”. The authors should elaborate on how they assessed robustness.**

This is also quite technical detail, which the reviewer rightly points out. We will like to iterate that the intended audience for this paper is that of clinicians and biologists. We have added a new section to the Supplement where we detail a robustness measure with regard to parameter ϵ . For each network we ran ϵ for values from 1 to 10 and compare values 1 – 9 to that of the network generated by $\epsilon = 10$. The overall mean value of recovery of edges is 99.36 % .(Please see Supplement for further details.) We also added to the main text (page 11):

More formally, we found that edges in the most sparse networks are consistently present in denser networks (99.3%, Supplementary Section 3).

- **The effect of parameter mu is shown in Suppl. Figure 1 for the AML dataset, some comment on whether similar effects apply on the other datasets would be useful.**

We have added to the supplement the plots for all other datasets to Figure 2, which display similar patterns but with reasonably distinct gradients.

Supplement Figure 2: Effect of μ on the number of variables for the four other datasets. From top left and travelling clock wise: MPN, myeloma, glioblastoma and colorectal.

- **Discussion of results would benefit from illustrating some key signalling events that were identified in the analysis and constitute novel observations.**

In all 5 datasets we have highlighted edges that we believe are novel in respect to the literature. The BNs are able to represent static dependencies between driver events at the time of sequencing. They have no dynamic component so they have no special power in representing upstream signalling events.

- **Indication of application to diseases other than cancer is mentioned**

in the manuscript, so the discussion could extend to some examples.

We added in the discussion (page), **Both in diseases with complex genomic elements that evolve over time (such as neurological diseases present predominantly on aging populations), but also in diseases where a number of non-genomic clinical variables can be used to build a network of dependencies (such as heart disease models based on lab and imaging variables).**

Minor comments: - “other BN algorithms of course can be used”, please rephrase to “of course other BN algorithms can be used”

fixed

- “Interested readers can find in these publications alternative BN learning algorithms and many more information in the nomenclature of BNs and their general use in computational biology”, please rephrase to “In these publications, interested readers can find alternative BN learning algorithms and more information on the nomenclature of BNs and their general use in computational biology”

fixed

- “as we seek for our methodology to be accessible to non-experts”, please rephrase to “as we seek that our methodology is accessible to non-experts”

fixed

- “We provide few parameter settings as to avoid reliance”, please rephrase to “We provide few parameter settings so as to avoid reliance”

fixed (many thanks for those)

Reviewer 2

The manuscript represents an attempt to develop and validate an AI based tool for a simple, network-based visualisation of big data variables and their relationships that would inform clinical and biological scenarios. The authors focus on generating networks of cancer genomic data from five different blood and solid cancers. The presented results largely vindicate their aims. However, there are several areas in each tumour type that need to be clarified further.

Dear reviewer many thanks for your time in reviewing our manuscript and insightful comments.

Methods 1. It is stated in the manuscript that the Bayesian networks are explainable AI models. What is the meaning of this and how does this become evident and verifiable in the case of the five different cancer patient cohorts studied here?

This is simply a descriptive statement of fact. BNs are (1) models, and (2) explainable. Our point in stating this fact is to juxtapose our method (A) to simple pairwise testing (which is routinely used in this context, and which are not models) and (B) AI models such as neural networks and deep learning which create models that are not interpretable.

We added to the end of page 4.

Comparing to pairwise methods BNs provide complex statistical models that intuitively presents the major relations within these type of data in one model. In contrast to neural networks and deep learning methods BNs provide visual cues to which relations between our variables are important.

2. in p10 para 3 it is stated ‘For instance all significant edges (shown as continuous lines) in Supplementary Figure 6 which shows the gBN for myeloma at $e = 12$ are also present at the much denser network for the same dataset shown in Figure 6 ($e = 3$). However, according to Suppl Fig 2, the density of edges anticorrelates with the e value, i.e., the density of networks in Suppl Fig 6 and Fig 6 should be the other way round.

Our statements are consistent with each other.

As you note $\epsilon = 12$ has the sparser network and $\epsilon = 3$ has the denser network. That is: higher ϵ leads to less edges so making the network sparser.

Please note that in reply to other comments, we have also added a new section to the Supplement (Section 3, Robustness) that will help clarify this and quantify the relation described in the text. In the new analysis that was added, we describe that in all five networks only a single edge was not consistently present across sparse and dense networks.

Added text to the manuscript (page 11):

More formally, we found that edges in the most sparse networks are consistently present in denser networks (99.3%, Supplementary Section 3).

3. p12-13 ‘For instance, for the family complex-NPM1-CEBPA bi in Figure 3 the AND gate is fitted between the edges complex-CEBPA bi and NPM1-CEBPA bi already identified by the BN learning algorithm (gated version is shown in Figure 4).’ The gate symbol shown in Fig 4 is OR not AND.

Fixed. Many thanks for spotting this.

Results AML It is a well established that t(15;17) can co-occur with FLT3-ITD in 20-30% of cases. This is not evident in Figures 3 &4?

In the Fig.3 there is a strong link from t(15;17) to NPM1, and from there to FLT3-ITD. So there is quite a direct path between t(15;17) and FLT3-ITD. In the context of our cohort the relation between t(15;17) and NPM1 appears to be very strong (as attested by the width of the edge), which might be an interesting hypothesis to investigate. The algorithm will prefer stronger relations as to build the best overall structure that explains the data.

Suppl Fig 15 is mentioned in p15 but not included in the Suppl File.

Added as Fig 11, with caption

Network for the AML dataset constructed by adding all edges for which the Fischer exact test (R’s `fisher.test()` function) between the two connected vertices returned a significant value (< 0.05).

MPN

If JAK2, CALR and MPL are independent early drivers, what is the meaning of MPL being the child and the other two the parents in the family?

1. The network and text state that JAK2, CALR and MPL are mutual exclusive to each other. So we make no claims about how early or late any of the drivers is. The mutual exclusivity of these 3 major players is an interesting finding that likely points to important biology.
2. There is no significance assigned to which drivers are children and those that are parents.
 - (A) That is why we draw networks without arrows.
 - (B) In order to establish directionality (which in some contexts can be interpreted as causality) one would need to run interventional experiments

We added to the manuscript (end of page 7):

In our networks no special interpretation should be given to the directionality of edges between family members. In particular, children-parents relations should be interpreted as correlations and not as causations.

Myeloma

NRAS and KRAS mutations are found in 40% of patients and in all cytogenetic subgroups, especially translocations. If my reading of the network is correct, the only conclusion that can be drawn from Fig 6 and Suppl Fig 5 is that NRAS and KRAS mutations are mutually exclusive to each other and to del13q. If so, then the network would be missing important clinical and biological information.

In our dataset (which is publically available at: https://github.com/nicos-angelopoulos/gbn/tree/main/data/gbns_in_cancer/mye) it is indeed the case that around 40% of patients have either NRAS or KRAS (42.54% to be precise), however only 22 patients have both mutations whereas 286 of them have either one or the other.

$$42.5414364640884 = (22 + 286)/724 * 100 \quad (1)$$

Our network therefore informs coincisely and precisely the appropriate information that there is a mutual exclusivity relation between NRAS and KRAS in this myeloma dataset.

We would be very happy to cite publications/datasets that have found different percentages.

We have added:

The network identifies mutual exclusivity between the MAPK pathway genes NRAS and KRAS. In our dataset 42% of the samples contain at least one of the two mutations with only 22 (3%) of the samples containing driver mutations to both genes.

In fact, in p20 it is stated that ‘Furthermore, novel patterns of co-occurrence were observed between other recurrent events such as between mutations on the MAPK pathway (NRAS and KRAS) and hyperdiploid cytogenetic status.’ Again, this co-occurrence is not evident in the network.

Our text was incorrect we have deleted the reference to NRAS and KRAS in that sentence. We added and changed text to read:

Furthermore, novel patterns of co-occurrence were observed between

other events such as between deletions of *CDKN2C* (*delCDKN2C*) and *FAM46C* (*delFAM46C*) and between *del13q14* and both *TRAF3* deletion (*delTRAF3*) and *(4;14)(MMSET;IGH)*.

Colon adenocarcinoma p21. ‘SOX9 mutations were mutated in a mutual exclusivity pattern to TP53 while significantly co-mutated with PCBP1 and MAP2K4.’ From the network it is evident that PCBP1 and MAP2K4 are co-mutated, the only link-edge for SOX9 denotes mutual exclusivity with TP3, itself in mutual exclusion with PCBP1. How does this pattern lead to the conclusion that SOX9 is co-mutated with PCBP1 and MAP2K4?

Our text was wrong, many thanks for spotting this. We have replace it with:

SOX9 was mutated in a mutual exclusivity pattern to TP53 while MAP2K4 significantly co-mutated with PCBP1, ATM and PTEN.

Suppl Fig 16 is missing.

Added as Fig 12 with caption

Network for the glioblastoma dataset constructed by adding all edges for which the Fischer exact test (R’s `fisher.test()` function) between the two connected vertices returned a significant value (< 0.05).

Reviewer 3

Overall I believe this paper is an interesting contribution to the field, it could help to popularise the use of an important technique into the field of cancer biology and I believe it could be published in this journal.

Dear reviewer thank you for your time in reviewing our manuscript and your positive comments.

It is really encouraging to see you comments, particularly with regard to the multi disciplinary appreciation to the work, which is often harder to come by.

REVIEWERS' COMMENTS:

Reviewer #1 (Remarks to the Author):

The authors have addressed my comments

Reviewer #2 (Remarks to the Author):

I am satisfied with the revised manuscript and am happy to recommend it for publication.

Reviewer #3 (Remarks to the Author):

In this revised version of the paper the authors clarify a number of issues brought up by others reviewers. Otherwise the paper is much the same and there are no huge changes based on the other reviewers comments.

My recommendations is the same as for the last version which is repeated below

Overall I believe this paper is an interesting contribution to the field, it could help to popularise the use of an important technique into the field of cancer biology and I believe it could be published in this journal.